# Fabrication and Performance Evaluation of a Novel Composite PVC-ZnO Membrane for Ciprofloxacin Removal by Polymer-Enhanced Ultrafiltration

**DOI:** 10.3390/polym16243551

**Published:** 2024-12-19

**Authors:** Sirisak Seansukato, Gangasalam Arthanareeswaran, Wirach Taweepreda

**Affiliations:** 1Polymer Science Program, Division of Physical Science, Faculty of Science, Prince of Songkla University, Hat-Yai 90110, Thailand; seansukato@gmail.com; 2Membrane Research Laboratory, Department of Chemical Engineering, National Institute of Technology, Tiruchirappalli 620 015, Tamil Nadu, India; arthanareeg@nitt.edu

**Keywords:** ciprofloxacin, zinc oxide membrane, polyvinyl chloride, ultrafiltration, polyvinyl alcohol

## Abstract

Water pollution is a major global issue, and antibiotic drugs released into aquatic environments by the pharmaceutical industry, such as ciprofloxacin, have negative consequences on both human health and the ecosystem. In this study, the performance of PVA as a polymer ligand for ciprofloxacin (CPFX) removal is evaluated through polymer-enhanced ultrafiltration using a novel composite PVC-ZnO membrane. The initial concentration of the ciprofloxacin solution, pH, ionic strength, ideal polymer concentration, duration, and maximum retention capacity were among the factors that were examined. In order to remove ciprofloxacin from water, PVA is utilized as a polymeric binding agent in a complex manufacturing process. In this instance, the PVC-ZnO membrane with 1.0 weight percent ZnO had a 96.77% ciprofloxacin clearance rate. PVA polymer has a high clearance rate of 99.98% in 1wt% of ZnO in this composite membrane when added to the ciprofloxacin solution. Fourier transform infrared spectroscopy (FTIR), energy dispersive X-ray spectroscopy (EDS), scanning electron microscopy (SEM), ultraviolet spectroscopy, and X-ray diffraction (XRD) were used to analyze the production and features of composite PVC-ZnO membranes. It is anticipated that this study’s discussion will be crucial to the development of higher-quality membrane technologies that remove pharmaceutical active chemicals from wastewater in an environmentally responsible manner without endangering the ecosystem. This investigation showed that composite PVC-ZnO membranes were effective materials for efficient removal of ciprofloxacin (CPFX).

## 1. Introduction

Due to the very limited fresh water in nature, many studies are focused on removing different types of pollutants from wastewater to produce clean water. Pharmaceutical compounds (e.g., antibiotics) are receiving attention and raising concerns because they pose threats to human lives and the environment. Their emergence has increased due to the growth in population, industrial activities, and climate change. These materials affect the quality of drinking water resources, e.g., spreading antibiotic (ciprofloxacin) resistance, and they can be toxic to aquatic organisms, which leads to loss of habitats and biodiversity. Moreover, their presence in water exceeding certain limits can cause adverse health problems such as brain damage, liver impairment, and carcinogenic diseases [1]. Ciprofloxacin’s (C_3_H_2_O_8_FN_3_O_3)_ molecular structure reveals the existence of functional groups that can bind to ion exchangers, specifically carboxylic acid groups. CPFX is a common medicinal substance that is utilized for a variety of medical conditions [1,2,3]. Humans and animals can utilize it, and it is often incompletely digested and expelled by urine and feces. As a result, many treatment technologies have been proposed to treat such types of wastewater, such as nanofiltration and ultrafiltration. Novel technology, like membrane technology, is desirable for wastewater treatment. Essentially, membrane separation processes have risen to be applied in water production, wastewater treatment [4,5,6], oil separation [7,8], etc. The polymeric membrane is more attractive due to the low cost, ease of fabricating, upscale, etc. In contrast, drawbacks such as hydrophobicity are the polymer’s nature, alongside its lesser mechanical and thermal properties than ceramics. There are various techniques to improve the hindrance of the polymeric membrane, such as the blending method, surface modification, and adding inorganic materials. The latter approach is straightforward and optimal for incrementing mechanical and thermal properties, enhancing hydrophilicity, and so on [9,10,11]. The polymeric membranes are generally made of some widely used materials like polyvinylidene fluoride (PVDF) [12], polysulfone (PSF) [13], polyether sulfone (PES) [14], cellulose acetate (CA) [15], and polyvinyl chloride (PVC) [4,16,17,18]. Among these materials, poly (vinyl chloride) PVC is desirable due to its versatile nature and variety of thermal, physical, and chemical properties. Another significant benefit of PVC is that it is used primarily for the inexpensive manufacturing of membrane technology. Recently, many researchers have been studying the preparation and characteristics of PVC and modified PVC membranes [6,18,19,20,21,22,23]. The membranes for this research are usually prepared via nonsolvent-induced phase separation (NIPS). PVC can be dissolved in various solvents such as N, N dimethyl acetamide (DMAc) [7,24], N-methyl-pyrrolidone (NMP) [5,10], tetrahydrofuran (THF) [9,25], and dimethylformamide (DMF) [15,19].

The performance of PVC membranes with the addition of different additives has been studied by Erdugan et al. [16]. The 0.5 wt% of ZnO NPs gave the highest water flux at pressure 0.7 bar. The compatibility between ZnO NPs and PVC-based membranes illustrated better filtration and mechanical properties than PSf-based membranes. Shaki et al. [26] studied the synthesis of PVC/ABS/ZnO nanocomposite films with zinc oxide nanoparticles (particle size less than 50 nm) and solution casting techniques. Nanocomposite films showed higher thermal stability than pure polymer. Antibacterial studied, the selective antibiofilm activity was studied against *Staphylococcus aureus* and *Pseudomonas aeruginosa*, PVC/ABS/ZnO nanocomposite showed larger zones of inhibition and higher antibiofilm and antioxidant activity than PVC/ABS polymer matrix. Sadek et al., [9] compared the metal oxide (i.e., Fe_2_O_3_, ZnO, and TiO_2)_ nanoparticles (NPs) mixing into nanocomposite thin films based on poly(vinyl chloride) (PVC) as a matrix. The XRD pattern and Raman spectra showed mixed ZnO metal oxide NPs in an amorphous region of PVC. SEM images revealed a homogeneous distribution of the NPs using 10.0 wt% in the PVC matrix, while the nanoparticle agglomerations at higher content (i.e., 15 wt%) were formed. As a function of temperature, the dynamic mechanical results indicated an increase in the storage and loss moduli and a decrease in loss factor (tan δ) with shifting to higher glass transition temperature, with increasing metal oxide NPs content up to 10 wt%. Thermal gravimetric analysis revealed that utilizing Fe_2_O_3_ and TiO_2_ NPs in the PVC matrix improved the thermal stability of PVC compared with ZnO NPs with catalytic behavior and polymer decomposition acceleration. Metal oxide NPs increased the electrical conductivity of PVC, and the electrical conductivity values were found to be in the order of 10^−13^ for ZnO. Other researchers reported the successful preparation and characterization of PVC and its composite membranes [4,15]. Alsalhy et al. [4] investigated the embedding of ZnO-NPs in the PVC membrane, resulting in a significant decrease in the CA value by 17.775°. The cake layer was reduced from 52.8 to 10.42 µm, and pure water permeability was enhanced by 315%. Asiri et al. [15] prepared an ultrafiltration ZnO-modified cellulose acetate membrane (CA/ZnO). The morphology of the membranes showed different pore sizes on rough surfaces and cross-sections, an asymmetric structure, and ultra-scale pores with an average pore radius of 0.0261 to 0.045 µm. ZnO-modified CA membranes have higher surface pore size, pore density, and porosity and improved surface hydrophilicity compared with pure CA membranes. Rabiee et al. [8] studied the enhanced thermal properties of incorporating ZnO NPs and improving the flux and antifouling properties of PVC/ZnO with ZnO NP.

CPFX is a common medicinal substance that is utilized for a variety of medical conditions. The predominant concentration range of CPFX found in water is between ng/L and μg/L. Regretfully, CPFX concentrations were shown to rise gradually over time, with a range of concentrations as high as mg/L [27]. CPFX has exhibited severe chemical interference and toxicity to microorganisms; it is also difficult for microbes to absorb and utilize CPFX. Numerous techniques, such as coagulation, membrane filtration, biological treatment or advanced oxidation processes, and electrochemical and adsorption processes, have been presented to remove emerging contaminants, like antibiotics, from water due to the growing demand for clean and safe water [28,29,30]. The simplicity of operation and relatively low energy consumption of membrane separation techniques make them very appealing solutions for the treatment of aqueous effluents [31,32]. Ultrafiltration (UF) is a commonly used membrane-based water and wastewater treatment technology. The intelligent design of UF membranes with antifouling potential maintains its high water permeability and removal efficiency. Although ultrafiltration membranes have pores with sizes ranging from 0.1 to 0.01 microns, one of the best ways to characterize UF membranes these days is molecular weight cut-off (MWCO). The molecular weight of a macromolecular solute at which 90% does not cross the membrane is known as its MWCO. High molecular weight and suspended particulates are trapped in what is known as the retentate, whereas low molecular weight and water solutes permeate the membrane as the permeate (filtrate) [3]. The retention of the water-soluble polymers by the ultrafiltration membrane influences the solute rejection. By employing PEUF, the functional water-soluble polymers with comparatively large molecular weights are utilized to capture all the tiny solute species via their interactions [33]. This process is called liquid-phase polymer-based retention (LPR) technology or polymer-enhanced ultrafiltration (PEUF). To separate low-molecular-weight species dispersed in an aqueous solution, the PEUF approach is a hybrid membrane separation technology that blends ultrafiltration membranes with water-soluble polymers [34]. Furthermore, ciprofloxacin (CPFX) can be effectively removed by the degradation of CPFX using (PEUF systems.

In order to assess the efficacy of the polymer retention technique for the removal of ciprofloxacin from aqueous effluents used as an initial step in the degradation processes, water-soluble polymers were tested for the most extraordinary interaction with various antibiotic species at different pH values [2]. The water-soluble polymer polyvinyl alcohol (PVA) exhibits the maximum removal of CPFX when filtering using a dead-end approach. Here, several of the composited PVC-ZnO membranes synthesized to remove the ciprofloxacin by using PEUF from water were evaluated and utilized.

## 2. Materials and Methods

### 2.1. Materials

PVC resin-medical grade powder (SCG-PVC SM61S) was supplied by SCG Chemicals Public Company Limited (Bangkok, Thailand), which is a homopolymer with a low molecular weight and K-value of 61 (corresponding degree of polymerization of 800). Dimethylformamide (DMF) with a purity of 99.8% was purchased from Sigma-Aldrich, St. Louis, MI, USA. ZnO nanoparticles supported by the College of Nanotechnology, King Mongkut’s Institute of Technology Ladkrabang, Bangkok, Thailand. Reverses osmosis (RO), deionized water, and other supports by the Department of Physical Science, Faculty of Science, Prince of Songkla University, Hat Yai, Songkla, Thailand.

### 2.2. Doped PVC Solution and Membrane Preparation

The PVC-ZnO composite membranes were fabricated using the phase inversion technique, which involves immersion precipitation in a nonsolvent coagulation bath. It was explained by earlier researchers [4,5,6,7,8]. Briefly, the flat sheet PVC and their nanocomposite membranes were fabricated by dissolved PVC resin powder (10–14 wt%) in a reagent glass bottle containing DMF according to Table 1, under continuous stirring by hotplate and magnetic stirrer at 60 °C for 2 h until a homogeneous solution was obtained. For the blended solution, zinc oxide nanoparticles (ZnO NPs) were dispersed into DMF under continuous stirring before dissolving PVC resin powder to prevent the aggregation of nanoparticles. PVC and blended solutions were degassed without stirring using an ultrasonic bath (sonicator) for at least 2 h to remove air bubbles and until air bubbles were no longer observed. Figure 1 illustrates the schematic of the neat and composite PVC membrane preparation; the glass plate was cleaned before casting. Pouring the dope solution on the flat surface of a glass plate and casting with a membrane thickness of about 200 μm under ambient temperature and controlled relative humidity of not more than 60%. The membrane films were immediately immersed into a coagulation bath with RO water; after they were peeled off from the glass surface, they were kept in RO water for at least 12 h until the residue solvent was eliminated.

### 2.3. Characterization of Zinc Oxide Nanoparticles

#### 2.3.1. X-Ray Diffraction (XRD)

The nanoparticles were covered with a thin layer of gold using the vacuum evaporation method to optimize the sample charging reaction due to the electron beam. X-ray diffraction spectrometer (XRD), Panalytical, Empyrean, Netherlands, equipped the Cu-Kα radiation (λ = 1.5406 Å) and used the operation voltages of 20 kV, Bragg’s diffraction angle in the range of 5–90° was applied to calculate the percentage of crystallinity and the estimated particle size of ZnO NPs is determined using Debye-Scherrer’s equation [35]:(1)t=0.9λ/βcos⁡θ
where t is the estimated average particle size, 0.9 is Scherrer’s constant, λ is the wavelength of the X-ray, which is equal to 0.15406 nm, β is the line broadening at full width at half maximum (FWHM) of the diffraction peak, and θ is Bragg’s diffraction angle.

The crystallinity index, CI(%) of the samples studied is calculated using the following equation:(2)$$CI(\%)=\left(S_{C}/S_{t}\right)\times 100$$
where $S_{C}$ represents the area of the crystalline domain and $S_{t}$ is the total domain area, respectively.

#### 2.3.2. Scanning Electron Microscope (SEM)

The morphology of ZnO nanoparticles, such as particle size and shape, was determined using a Scanning Electron Microscope (SEM), Hitachi, SU3900, Japan, with an operating voltage of 20 kV and 25,000 times magnification.

### 2.4. Membrane Characterization

#### 2.4.1. Fourier Transform Infrared (FT-IR) Spectroscopy

The FTIR spectra were performed using a Bruker Tensor 27 (Bruker Optics GmbH, Karlsruhe, Germany. The FT-IR spectrometer is equipped with ATR mode from Pike Technologies GladiATR (Bruker Optics GmbH, Karlsruhe, Germany). During measurements, the sample cell (proves) was cleaned with ethyl alcohol. The FT-IR spectra of membrane samples, namely P(10-14)-(0-2)ZP, were observed with a scan range of 400–4000 cm^−1^ at a resolution of 64 scans.

#### 2.4.2. Pore Size and Porosity

The average porosity of the fabricated membranes was measured by the gravimetric method, as indicated below [6,34].
(3)$$P=\frac{\left(w_{1}-w_{2}\right)}{A\times l\times \rho_{w}}$$
where $w_{1}$ and $w_{2}$ are the weights of the wet and dry membranes, respectively. A is the membrane area (m^2^), l is the membrane thickness (m), and $\rho_{w}$ is the water density (0.998 g/cm^3^). For porosity measurement, at first, pieces of the membrane with specific areas were immersed in distilled water for at least 12 h to ensure that all the pores of the membranes were filled. Immediately after that, water on the surface of the samples was cleaned cautiously and weighed. Subsequently, the samples were put in an oven for 2 h at 60 °C to evaporate water from membrane pores and weighed again.

Guerout–Elford–Ferry relation (Equation (4)) was used to measure the mean pore radius $\left(r_{m}\right)$ of the membranes [6,34]:(4)$$r_{m}=\sqrt{\frac{\left(2.9-1.75\varepsilon \right)\times 8\eta lQ}{\varepsilon \times A\times ∆P}}$$
where ε is the overall porosity, η is the water viscosity $\left(8.9\times 10^{-4}\;Pa.s\right)$, l is the wet-membrane thickness (m), A is the membrane surface area (m^2^), Q is water flux (m^3^/s), and ∆P is the operation pressure (0.2 MPa).

#### 2.4.3. Scanning Electron Microscope (SEM)

The surface and cross-section morphologies of neat and nanocomposite membranes were observed by scanning electron microscope (SEM) with equipped EDX using HITACHI, SU3900, Japan, operating at 20 kV accelerating voltage. Prior to imaging, the surface of all samples was placed on a sample holder and coated with a thin layer of gold. SEM images of the upper and lower surfaces of all flat sheet membrane samples were coated with a thin layer of gold. They were immersed in liquid nitrogen for cross-section images to preserve the microstructure and fracture with metal pliers. The energy-dispersive X-ray mapping technique analyzed element distribution with a 20 kV operating voltage.

Energy-dispersive X-ray spectroscopy (EDS, EDX, EDXS, or XEDS), also called energy-dispersive X-ray analysis (EDXA) or energy-dispersive X-ray microanalysis (EDXMA), is an analytical technique used for the elemental analysis or chemical characterization of a sample.

The interpretation of EDX with an example is given below:Weight% = 100 x weight of one component (element)/weight of the entire sample.
Atomic% = 100 x number of atoms of one component/total number of all atoms in the sample.

To convert from weight percentage to atomic percentage, use the mole fraction of each component divided by the total mole of all components as the following Equation:(5)$$at\;\%\;A=\frac{mole\;A}{mole\;A+mole\;B}\times 100$$

It can be used in another form to determine whether using the following equation:(6)$$(at\;\%)\;A=\frac{W_{A}\;A_{B}^{\prime }}{W_{B}A_{A}^{\prime }+W_{A}A_{B}^{\prime }}\times 100$$

This equation converts the weight ratio of each chemical composition from the EDX spectrum data to the atomic ratio, in which the atomic percentage of the presence of an additional filler ratio in the PVC-composite membranes will be discussed.

#### 2.4.4. Mechanical Properties

Mechanical properties were evaluated by a Universal Testing Machine (UTM) from Lloyd, TA plus, Germany. The sample dimension was cut into rectangular shapes of 1 cm × 5 cm, a gauge length of 4 cm, and a tension speed of 20 mm/min; at least three reputations (*n* = 3) per condition were collected and evaluated via statistics of the Least Significant Different, LSD method. Stress–strain curves, ultimate tensile strength, and Young’s modulus were determined for the mechanical properties of the PVC-based composite membranes.

#### 2.4.5. Contact Angle Measurement

The contact angle measurements for membrane samples were performed with an optical contact angle measurement (OCA 15 EC) from Dataphysics Instruments GmbH. The area of the samples was at least 5 cm^2^ (sample width and length are 1 cm × 5 cm). To measure each sample, a 15 μL droplet of deionized (DI) water was placed from the tip of a syringe to the membrane surface. When the water drop contacted the surface of the membrane film, the contact angle was measured at different positions and times.

#### 2.4.6. Pure Water Flux

Flat sheet membranes of all conditions were stored in distilled water for 12 h before being brought to observe the water permeability using a lab scale, dead-end mode water filtration apparatus. Set up the membranes into the dead-end module, adjusting the hydraulic pressure from 0–5 bar and operating time from 0 to 30 min, as shown in Figure 2. Recording the permeation flux in two methods. Firstly, record the permeated pure water flux using the balance in each time interval from 0, 1, 2, 3, 5, 7, 10, 20, and 30 min at a hydraulic pressure of 1 bar. Plot graph of permeated flux versus time interval. Secondly, the time interval of 30 min should be fixed, and the operating pressure should be adjusted from 0 to 5 bar, which measures each pressure level. Plot the relationship between the permeation of pure water flux and operating pressure [5]. The slope of this curve was calculated to the water permeability of the neat PVC and nanocomposite membranes. It is calculated by
(7)$$J=\frac{V}{∆t\times A}$$

Here, *J* is known as the water flux (L/m^2^ h), *V* is the volume of the permeate solution (*l*), ∆t is the time difference of permeate flux (h), and *A* is the area of the membrane (m^2^).

#### 2.4.7. Preparation of a Ciprofloxacin Solution

Next, 10 mg/L of ciprofloxacin was mixed with distilled water, and the solution was stirred for 10 to 20 min. Ciprofloxacin with PVA was used as a temporary binder owing to its water solubility, excellent binding strength, and clean burning characteristics. Four grams of PVA was added to 100 mL of distilled water to prepare the PVA solution. The mixture was stirred at 500 rpm until the PVA was fully soluble in the water to form a homogenous solution. Then, ciprofloxacin was added, and the solution was brought to a volume of 1 L by adding distilled water.

Then, the rejection of ciprofloxacin (%) is calculated by Equation (8),
(8)$$R\left(\%\right)=\left(1-\frac{C_{p}}{C_{f}}\right)\times 100$$

R is known as the rejection; $C_{p}$ is the concentration of the permeate and $C_{f}$ is the concentration of the feed solution.

## 3. Results and Discussion

### 3.1. Zinc Oxide Characterizations

The crystallinity, particle size, and shape of ZnO nanoparticles are evaluated using the X-ray Diffraction (XRD) technique; the corresponding XRD pattern is shown in Figure 3a. Zinc oxide nanoparticles represent eight distinct peaks at 2θ = 31.7°, 34.4°, 36.2°, 47.5°, 56.5°, 62.8°, 67.9°, and 69.0° as corresponding to standard Bragg reflection planes (100), (002), (101), (102), (110), (103), (112) and (201). Moreover, ZnO NPs structure was identified as a hexagonal wurtzite structure [36,37,38] with lattice parameters of a = b = 3.2522 Å and c = 5.2095 Å (Reference ICCD code number: 01-078-2585). The average particle size of ZnO NPs is 6.53 nm due to the FWHM of the most intense peak located at 36.2° (101) using Equation (1). The crystallinity index of ZnO NPs was obtained following Equation (2), which shows the calculated crystallinity of ZnO NPs is 91.41%.

The morphology of ZnO nanoparticles, such as particle size and shape, was observed using a scanning electron microscope (SEM). Figure 3b shows that the nano−ZnO particles present in a form like a hexagonal rod exhibit the aggregation of nanoparticles. The particle size of ZnO nanoparticles exhibits various sizes from 125 to 135 nm, measured in lateral diameter using ImageJ 1.53 software and plotted with Origin. However, the shape of the particles is hexagonal rods at a magnification of 25,000 times.

### 3.2. The PVC-Based Membrane Characterizations

#### 3.2.1. Fourier Transform Infrared (FT-IR) Spectroscopy

The pristine PVC and ZnO chemical structure can be seen in Figure 4a. For pure PVC (red line), the absorption band is observed at 2970 and 2911 cm^−1^, which are assigned to asymmetric C–H stretching bonds. The absorption band located at 1427 cm^−1^ is assigned to symmetric C–H stretching bonds. The vibration peak located at 1253 cm^−1^ is assigned to the C–H bending bond. A peak located at 1096 cm^−1^ is assigned to C–C stretching bonds. The 961 and 613 cm^−1^ peaks contributed to the C–Cl stretching and C–Cl bending bonds, respectively. The spectrum of the ZnO particle (green line) presented the main absorption band located at 400–600 cm^−1^, which is attributed to the Zn–O stretching bond. These experimental data of wavenumbers correspond to the other researchers [19,39,40,41,42].

The FTIR spectra of the PVC/ZnO nanocomposite membranes with the difference concentrations of PVC (10–14 wt%) and ZnO particle loadings (0.1–1.0 wt%) are shown in Figure 4b. The absorption bands of PVC/ZnO nanocomposite membranes are observed at 3100–3500 cm^−1^ assigned to the O–H stretching bonds, and the intensity of this absorbance range increases with the increase of ZnO loading. The prominent absorption bands involved in PVC structure, such as asymmetric and symmetric C–H stretching (2911–2970 cm^−1^ and 1427 cm^−1^), C–H bending (1096 cm^−1^), C–Cl stretching and bending bonds (960 cm^−1^ and 610 cm^−1)^ are corresponding to the PVC spectrum in Figure 4a. Meanwhile, the absorption bands at 400–600 cm^−1^ are attributed to the Zn–O stretching bond. This absorption band can be cleared so that the intensity of the Zn–O stretching bond increases with the increment of ZnO loading. Furthermore, the weak peak at 1651 cm^−1^ is attributed to the O–H bending bonds due to the complexation between ZnO particles and PVC backbones [40]. Finally, increasing Zn–O stretching bonds could increase the polymeric structure’s hydroxyl group (–OH groups). These results involve the improvement of the hydrophilic of the PVC-ZnO composite membranes, which results in morphological structure, porosity, water uptake, and mean pore size of the membranes.

#### 3.2.2. Morphology and Energy X-Ray Dispersive Spectroscopy Analysis

EDX analysis was performed to confirm the formation of ZnO particles into the PVC-based flat sheet membranes. During the EDX measurement, various areas were focused, and the corresponding peaks are shown in Figure 5, Figure 6, Figure 7, Figure 8, Figure 9 and Figure 10. Firstly, for the nascent PVC membrane, as in Figure 5 and Figure 6, the chemical components such as carbon, chlorine, and oxygen can be seen in the prepared PVC membranes in the EDX spectra. In Spectrum 1, the quantity on the surface area by atomic percent of C, Cl, and O were 90.31, 8.12, and 1.57, respectively. At the same time, the quantities at the cross-section of the membrane were 85.55, 10.78, and 3.67 (measured in atomic % for C, Cl, and O, respectively).

Secondly, as in Figure 7, Figure 8, Figure 9 and Figure 10 and Table 2, the PVC-composite membranes illustrated the elements of carbon, chlorine, oxygen, and zinc in the prepared MMMs by adding 1.0 wt% of ZnO particles in the EDX spectra. The ZnO particles were almost more homogeneously dispersed in the surface area than in the cross-section. In contrast, the cross-section images had the aggregation of ZnO particles.

The morphology of PVC-composite membranes’ upper and lower surfaces was illustrated through SEM images, as shown in Figure 11 and Figure 12. At a magnification of 20,000 times, the upper surface displayed micropores distributed randomly on the entire surface. Micropores of 10 wt% are more significant than 14 wt%. On the other hand, at 5000 times the magnification of the lower surface of 10 wt%, the composited membrane has micropores in the range of size from 0.5 to 2.5 μm, distributed through the entire lower surface, and has more than 14 wt% MMM.

From a flat sheet membrane’s cross-section SEM images (Figure 13 and Figure 14), 14 wt% PVC -1.0 wt% ZnO NPs display the sponge-like structure on the entire bulk membrane, except the top layer has finger-like pores or micropores. Moreover, the surface of voids at the bottom zone would show the micropores for almost all the holes except for the top site. These micropores still occur at the surface of the finger-like pore at the top layer, but there are fewer than at the voids in both pore amount and size. The embedded ZnO NPs are helpful in increasing the pore size in the middle layer of the composite membranes, resulting in an increase in the mean pore size at higher ZnO concentrations.

#### 3.2.3. Porosity and Water Uptake

Link to SEM information: From Table 3, The porosity of pristine PVC membranes (12–14 wt%) increased up to 80% with the addition of 0.1 wt% ZnO to the casting solution, except pristine PVC 10 wt% increased to 76% with the incorporation of 0.5 wt% ZnO to dope solution. In contrast, the average pore size of PVC-based composite (ZnO) membrane increased to 24% with the addition of 1.0 wt% ZnO to the dope solution in all concentrations of PVC. Adding hydrophilic nanoparticles into the casting solution facilitates the demixing process between solvent and nonsolvent, increasing water transport into the composite membrane and resulting in a higher porosity and surface pore size. The increment in loadings of ZnO beyond 0.1 wt.% resulted in a reduction in porosity and an increase in mean pore size, probably due to the accumulation of nanoparticles [15]. Furthermore, these behaviors can be attributed to the increased casting solution viscosity with higher nanoparticle loading, which hinders and slows the phase inversion process between solvent and nonsolvent.

#### 3.2.4. Hydrophilicity

Wettability is defined as the ability of the surface wetting between liquid and solid phases to react with each other. Contact angle measurements are the most common method to evaluate the order of wettability, especially for polymeric membrane surfaces. Generally, the lower contact angle (Ø < 90°) is referred to as higher wettability or more hydrophilicity, which can be explained by the surface of the modified membrane, like the water, and can absorb water molecules passing through itself. While the higher contact angle (Ø > 90°) is defined as lower wettability or more hydrophobicity, in this case, the surface of the modified membrane does not like water and cannot absorb water.

Figure 15 reveals the contact angle of the neat 10–14 wt% PVC membrane and their nanocomposite membranes with 0.1–1.0 wt% of ZnO NPs, which indicates that incorporating ZnO NPs increases the surface hydrophilicity of the modified PVC membranes. The pristine PVC membranes have an average contact angle of 89–91°; this illustrates the natural hydrophobic behavior of PVC. When adding 0.1 wt% of ZnO NPs, the contact angle decreases by 10°, and the wettability of the modified PVC membrane becomes more hydrophilic compared to the virgin one. For the addition of ZnO NPs over 0.1 wt%, the hydrophilicity of the composite membranes was slightly decreased, and they have higher hydrophilicity than the pristine. The aggregation of ZnO particles at higher concentrations affected the dispersion of the surface of the membranes, and the agglomerated particles precipitated on the bottom surface of the membranes. Corresponding to Alsalhy, F.Q. et al. (2018) [4] evaluated the effect of the addition of ZnO NPs into the PVC membrane with a small ratio of 0.3 g; the contact angle was reduced by 17.775° from the neat 13 wt% PVC membrane of 64.005°. However, the contact angle of the nanocomposite membranes increases slightly with the increase in the ZnO NPs content up to 1.0 wt%. The initial phenomenon of the significant decrease of the contact angle shows more hydrophilicity of the nanocomposite membrane due to the nanoparticles moving toward the surface of the membrane and because of the hydrophilic of nanoparticles, resulting in increased hydrophilic behavior of the modified membranes. With the higher ratio of nanoparticles, the hydrophilicity of the modified membranes becomes less than the small ratio due to the aggregation and lower dispersal of nanoparticles.

#### 3.2.5. Mechanical Properties

The mechanical properties of the different PVC-based membrane concentrations, consisting of tensile strength, Young’s modulus, load at break, stress at break, and strain at break, were measured using the universal testing machine, LLOYD instrument, TA plus, U.S.A. The statistical analysis in this section is the data collected from the measurement of three reputations (*n* = 3) using data analysis in Microsoft Excel with the Anova single factor and given an alpha value of 0.05. Then, the mean ± standard deviation was presented and compared to each condition using the LSD method.

The flat sheet membrane was obtained by casting on the glass plate and using the phase inversion technique. The sample preparation is cut into a rectangular shape, and the testing condition is operated using the ASTM D412 method. Table 4 shows that the tensile strength and load at break of 10 wt% PVC-based were decreased with the addition of ZnO particles compared to the virgin. Young’s modulus and strain at break increased as the ZnO ratio increased to 0.5 wt%, whereas adding ZnO content at 1.0 wt% decreased both Young’s modulus and strain at break. Fluctuating ranges from 3.37 to 3.59 MPa of stress at break data occurred with the addition of ZnO particles.

Table 5 shows that the tensile strength, Young’s modulus, load at break, and stress at break of 12 wt% PVC-based were significantly decreased by adding a starting ratio of 0.1 wt% of ZnO particles compared to the virgin. These properties increased with the increment of ZnO particles up to 1.0 wt%. This condition only strains at the break, alternating up and down with adding ZnO particles.

Table 6 shows that the tensile strength of 14 wt% PVC-based membranes was significantly decreased when ZnO particles were added. Young’s modulus increased dramatically with the increment of ZnO particles up to 0.5 wt% and then fell at 1.0 wt%. Load at break and stress at break were improved by adding a starting ratio of 0.1 wt% of ZnO particles; the increase of ZnO particles significantly decreased load at break and stress at break. Strain at break was reduced considerably with the increment of ZnO particles up to 0.5 wt and then increased with the addition of ZnO particles of 1.0 wt%.

The comparison of the mechanical properties of PVC-based membranes with the difference in PVC concentrations and ZnO particle contents was illustrated in Figure 16, Figure 17, Figure 18, Figure 19 and Figure 20. The tensile strength displayed in Figure 16 shows the higher PVC concentrations increased tensile strength compared to the low concentration of 10 wt% in nascent and inorganic composite membranes. Thus, incorporating ZnO particles in the PVC-based membranes decreased the tensile strength compared to the neat PVC membrane. In Figure 17, Young’s modulus, the lowest at 10.0 wt% PVC ratio, was quite different with the increase of ZnO particles. In comparison, Young’s modulus of 12 wt% PVC membranes decreased with the growth of ZnO particles up to 0.5 wt% and became higher with a 1.0 wt% ZnO content. Interestingly, Young’s modulus of 14 wt% PVC membranes was increased with the increment of ZnO content. 

Load at break of PVC-based membranes with the difference in PVC concentration and ZnO content was demonstrated in Figure 18. As can be seen, the PVC concentration strongly affected the change in load at the break of the membranes; the 10.0 wt% PVC ratio was slightly changed in load at the break. On the other hand, the increase of ZnO content in a 10.0 wt% PVC-based membrane was unchanged in load at the break. Meanwhile, the incorporation of 0.1 wt% ZnO content in 12 wt% PVC membrane decreased load at break enormously significantly, and after increasing ZnO content up to 1.0 wt%, load at break increased. Adding 0.1 wt% ZnO ratio into 14 wt% PVC membrane increases the load at the break, and then the load at the break of 14 wt% PVC membrane was significantly decreased with the increase of ZnO ratio up to 1.0 wt%.

The stress at break of PVC-based membranes with the different PVC and ZnO particle concentrations, as seen in Figure 19, showed that the higher PVC concentration significantly increased the stress at break. In comparison, the ZnO content decreased significantly of stress at break in the range of 0.1 to 1.0 wt% of ZnO content, and the higher PVC concentration, except for the 10.0 wt% PVC membrane with any ratio of ZnO particles, was not changed in stress at break.

The strain at break of PVC-based membranes with the different PVC and ZnO particle concentrations as can be seen in Figure 20, both difference PVC and ZnO concentrations would not be affected by the strain at break, except at 0.5 wt% of ZnO content, the strain at break decreased significantly with the increment of PVC concentration.

#### 3.2.6. Pure Water Permeability of Neat and Composite PVC-ZnO Membranes

The Pure Water Flux PWF of the neat and composite PVC-ZnO membranes is illustrated in Figure 21**.** The results indicate the decreasing trend of PWF with the inclusion of nanomaterials. The water flux in the PVC-ZnO membrane was 34.46 ± 1.3 L m^−2^ h^−1^, whereas all the other composite ZnO membranes showed significantly lower fluxes than the PVC membrane. The water flux of N10-0.1ZP membranes was noted to be 20.8 ± 1.6 m^−2^ h^−1^, while N12-0.1ZP and N14-0.1ZP membranes showed 4.18 ± 0.01 L m^−2^ h^−1^ and 0.19 ± 0.2 L m^−2^ h^−1^. The PWF of N12-0.1ZP and N14-0.1ZP membranes were 77.78% and 70.37% higher than the N10-0.1ZP membrane, which is a noticeable decrease in water fluxes, indicating the influence of the ZnO materials on the membrane permeation behavior mainly in two ways. Firstly, it enhances the hydrophilicity of the PVC-ZnO membranes, and secondly, it impacts membrane morphology. The decrease in porous finger-like structure was observed from the cross-sectional morphology of nanomaterials incorporated in ZnO membranes. Furthermore, the rapid phase demixing by the presence of (0.1, 0.5, 1.0) ZnO nanomaterials in the casting solution may decrease the thermodynamic instability, resulting in a lower porosity in the membrane skin layer, ending up in a decrease in the membrane fluxes. The massive macro voids were observed in the PVC-incorporated membrane due to its affinity towards polymer. This decreased the interaction and mass transfer rate between the membrane surface and feed solution during the separation performance compared to the neat PVC-ZnO membrane. The enhanced hydrophilicity with nanomaterials could not attract water molecules in the membrane matrix, prevent them from passing through the membrane structure, and significantly decrease permeability [43,44]. In addition, at higher ZnO concentrations, there is a tendency for more aggregation, causing the decreasing flux due to the clog on the membrane pores.

#### 3.2.7. Ciprofloxacin (CPFX) Removal by PVC-ZnO Membrane Between Mixing PVA Polymer Without Mixing PVA Polymer

##### Ciprofloxacin (CPFX) removal is achieved by using a PVC-ZnO membrane without mixing the PVA polymer

Determination of the effect of ciprofloxacin retention by PVC-ZnO composite membrane with a fixed concentration of 10 ppm with a constant pressure at 4 bar. The result is depicted in Figure 22. This shows the gradual decrease in the CPFX flux with increasing concentration of PVC-ZnO composites membrane. The flux decreases with increasing concentration, but the retention increases with decreasing flux. This results in low concentration, huge pore size, and low absorbance. This leads to reduced flux (18.46–0.0642) L/m^2^ h for CPFX. Furthermore, the decrease in flux in this experiment is due to the aggregation of ZnO NPs at higher concentrations compared to the pristine PVC membranes, as explained in a later section on pure water permeability.

##### Ciprofloxacin rejection of a nonPVA polymer

Because of its solubility and durability, ciprofloxacin is a small-molecule antibiotic that can be difficult to extract from water efficiently. Ciprofloxacin rejection percentage might also be influenced by particular ultrafiltration parameters (pH, pressure, concentration, etc.). The efficacy of removing Ciprofloxacin (CPFX) from a synthetic ciprofloxacin solution was examined in a procedure devised to produce a composite PVC-ZnO membrane [45]. With a constant operating pressure of 4 bar, the impact of a 10 ppm starting concentration of ciprofloxacin on rejection efficiency and permeating flux was investigated, as shown in Figure 23. The necessary times for the permeate flow and rejection efficiency were also noted. The constructed PVC-ZnO membrane produced positive outcomes. The highest percentage of measured ciprofloxacin rejection was 96.77%, shown in Figure 24. The work’s outcomes demonstrated how these membranes may be used to remove ciprofloxacin efficiently [1].

##### Ciprofloxacin (CPFX) flux by the PVC-ZnO membrane with a PVA polymer

A low amount of ciprofloxacin molecules are adsorbed onto the surface of the PVC-ZnO composite membrane through low non-covalent interactions such as van der Waals forces, hydrogen bonding, or electrostatic interactions. Therefore, we are mixing water-soluble polymer (PVA) with the ciprofloxacin solution within the same concentration to increase the molecular size [24]. The ciprofloxacin molecules may form complexes with functional groups present in the PVA polymer [46], leading to the binding of the drug to the polymer. This leads to a decrease in flux (Figure 25) and an increase in the rejection percentage of CPFX. Here, the Polymer Enhanced Ultrafiltration (PEUF) decreases the ciprofloxacin flux (16.46-0.0842) L/m^2^ h.

##### Ciprofloxacin rejection with PVA polymer

Rejection of CPFX was carried out batch-wise in a dead-end cell. Initially, CPFX solutions were prepared with the addition of PVA to check the rejection rate through PVC-ZnO composite membranes, then collected samples were analyzed for their absorbance using a UV–Vis spectrometer to check the rejection percentage. After adding PVA to the feed solution, 99.99% rejection was observed in all CPFX with PVA solutions—the rejection aptitude of the CPFX to the concentration of ZnO. The concentration of the CPFX in permeate was analyzed using a UV–Vis spectrophotometer. An appropriate calibration curve determines concentration. The percentage of rejection is calculated using the formula,
$$R\left(\%\right)=\left(1-\frac{C_{p}}{C_{f}}\right)\times 10$$
where $C_{p}$ is the initial concentration of the solution, $C_{f}$ is the final concentration of the solution. The experiment was carried out on all CPFX solutions with different concentrations of ZnO-composite membranes. The 1.0 wt % ZnO loaded membrane demonstrated 99.99% CPFX rejection, shown in Figure 26; however, the membranes having 0.1% and 0.5% ZnO had pinholes because of excess hydrophilic fillers, which leads to lesser CPFX rejection. This observation is further supported by the data obtained from water flux studies by UV–Vis spectrophotometer. The hydrophilicity and steric hindrance of the membrane plays a vital role in the CPFX ejection. The 0.1% ZnO membrane is less hydrophilic, so the interaction between water on the membrane surface is less; therefore, the CPFX rejection is decreased. The 1.0% ZnO-loaded membrane exhibits more significant rejection than the remaining fabricated membranes due to the steric exclusion of the CPFX during the pore entry [47]. In this study, the adsorption process plays a negligible role because this membrane is found to exhibit more excellent rejection for both anionic and cationic CPFX, and the other reason is that an increase in ZnO-loaded membranes showed a decrease in rejection. Hence, it confirms that the rejection phenomenon occurs due to hydrophilicity and pore size. The rejection of ciprofloxacin removal is shown in Figure 27.

## 4. Conclusions

This study evaluated the composite PVC-ZnO membranes synthesized to remove the ciprofloxacin by using PEUF from water. The results indicate the decreasing trend of PWF by including ZnO materials in the PVC membrane matrix. The ciprofloxacin molecules were complexes with PVA polymer, leading to the binding and enhancing the solute size. This leads to a decrease in flux and an increase in the rejection percentage of CPFX. In this instance, the rejection rate was 96.77% without the PVA polymer in the ciprofloxacin solution; however, it rises to 99.98% when the PVA polymer is mixed with the ciprofloxacin solution. The characteristics of composite PVC-ZnO membranes showed better membrane performance by incorporating ZnO, which is the benefit of membrane filtration techniques. This experiment further validates that using composite PVC-ZnO membranes is a promising and efficient technique for eliminating pharmaceutical waste. Future research investigations should focus on evaluating such membranes for the selective removal of ciprofloxacin and investigating the membrane’s performance in the presence of other pollutants.

## Figures and Tables

**Figure 1 polymers-16-03551-f001:**
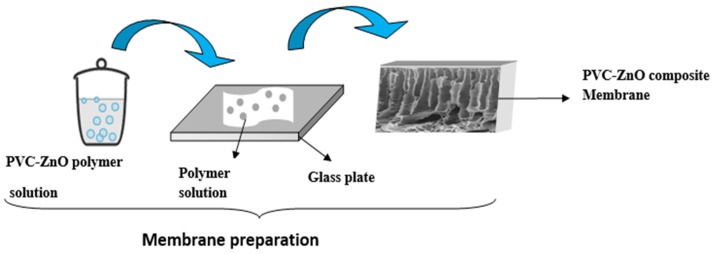
Schematic of flat sheet neat and composite PVC-ZnO membranes preparation.

**Figure 2 polymers-16-03551-f002:**
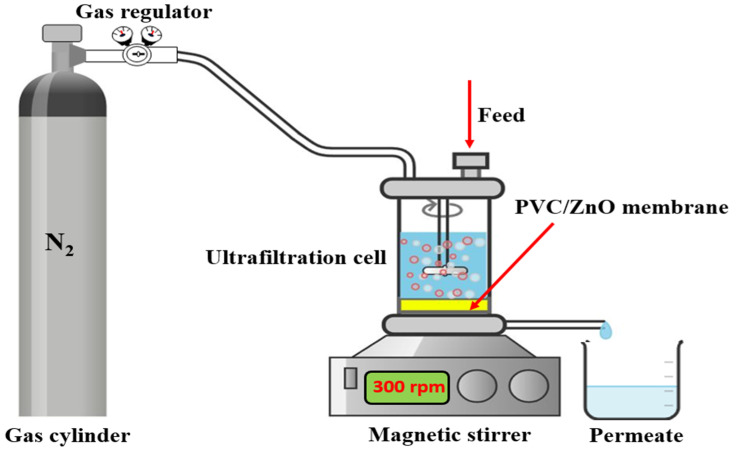
Schematic diagram of the ultrafiltration setup.

**Figure 3 polymers-16-03551-f003:**
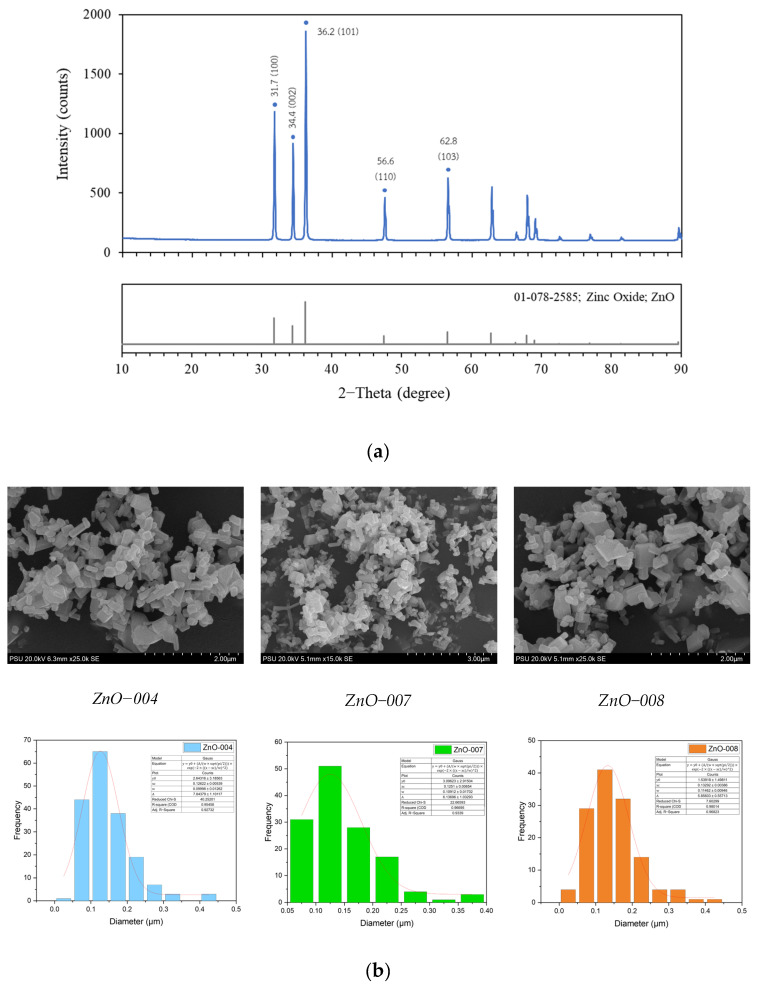
ZnO particle characterizations (**a**) XRD spectrum and (**b**) SEM image and particle size distribution.

**Figure 4 polymers-16-03551-f004:**
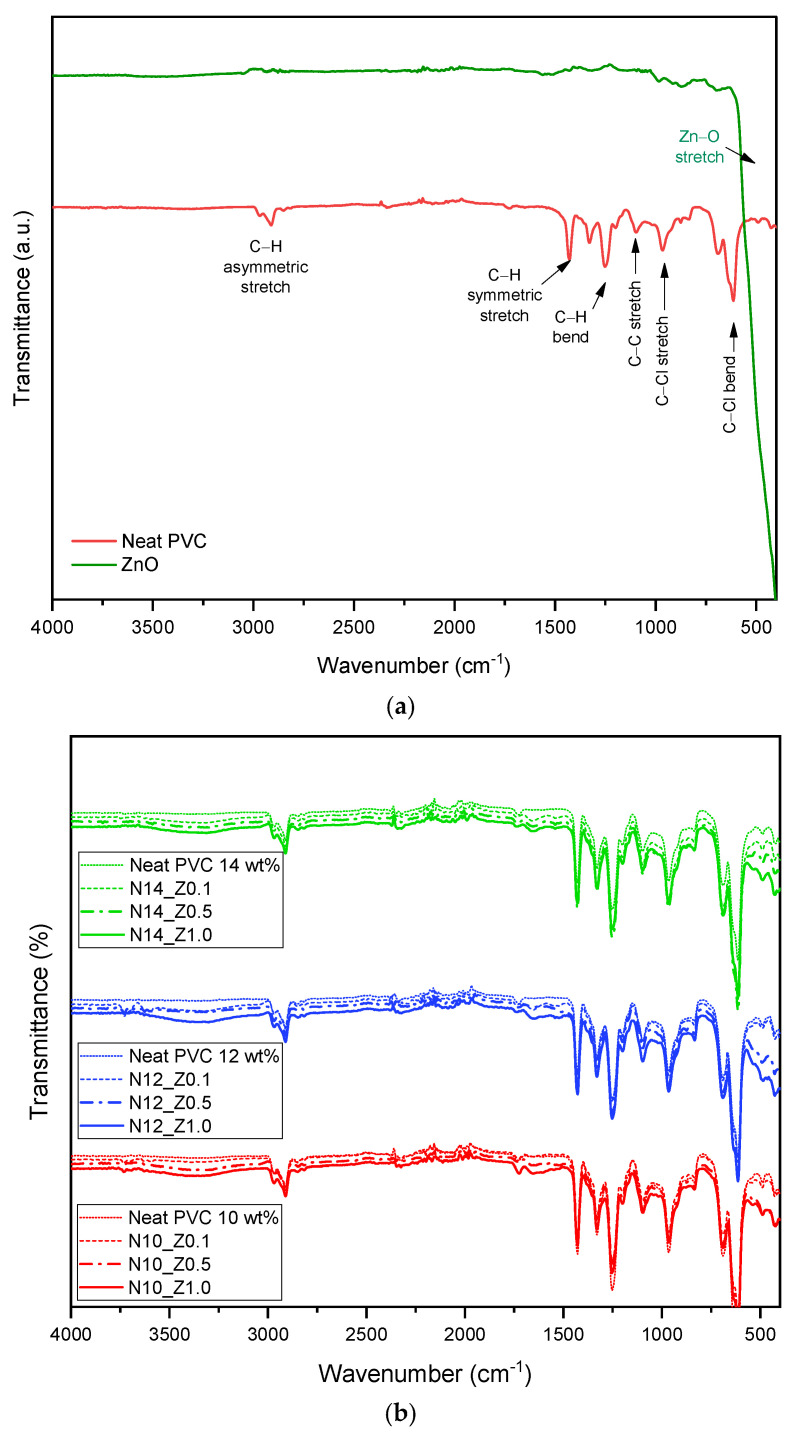
FTIR spectra of (**a**) pristine PVC and pure ZnO particles and (**b**) nanocomposite PVC membranes.

**Figure 5 polymers-16-03551-f005:**
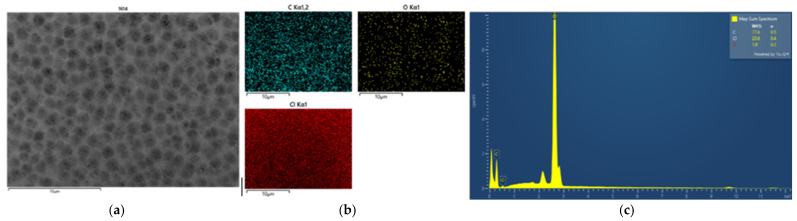
SEM image of pure PVC 14 wt% (**a**) top surface, (**b**) mapping images, and (**c**) element spectrum.

**Figure 6 polymers-16-03551-f006:**
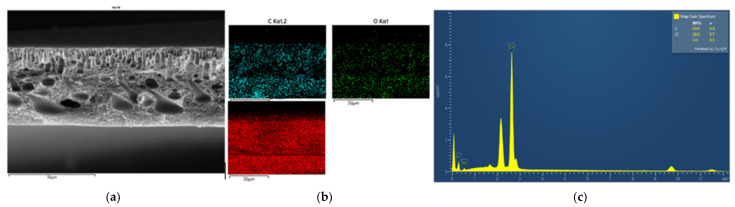
SEM image of pure PVC 14 wt% (**a**) cross-section, (**b**) mapping images, and (**c**) element spectrum.

**Figure 7 polymers-16-03551-f007:**
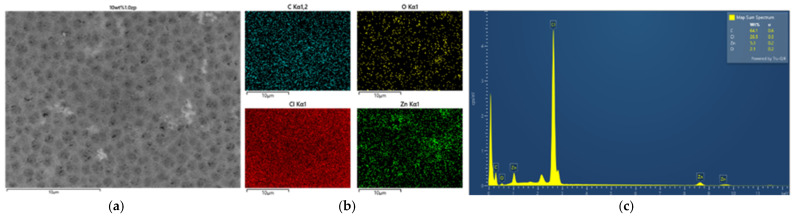
SEM image of PVC 10 wt% and 1.0 wt% of ZnO (**a**) top surface, (**b**) mapping images, and (**c**) element spectrum.

**Figure 8 polymers-16-03551-f008:**
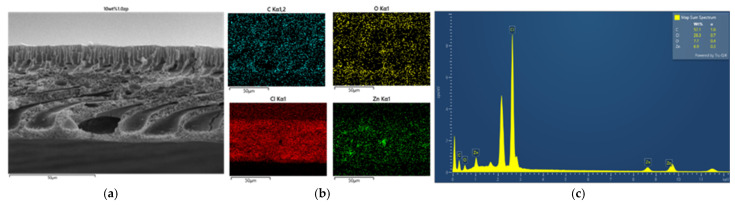
SEM image of PVC 10 wt% and 1.0 wt% of ZnO (**a**) cross-section, (**b**) mapping images, and (**c**) element spectrum.

**Figure 9 polymers-16-03551-f009:**
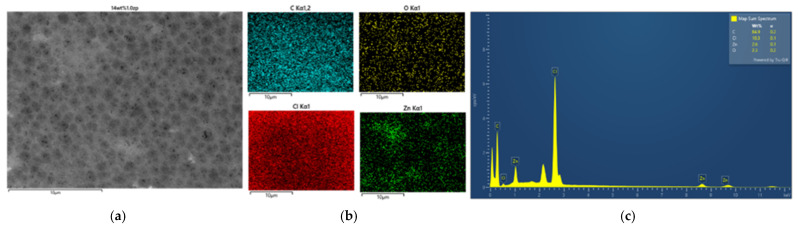
SEM image of PVC 14 wt% and 1.0 wt% of ZnO (**a**) top surface, (**b**) mapping images, and (**c**) element spectrum.

**Figure 10 polymers-16-03551-f010:**
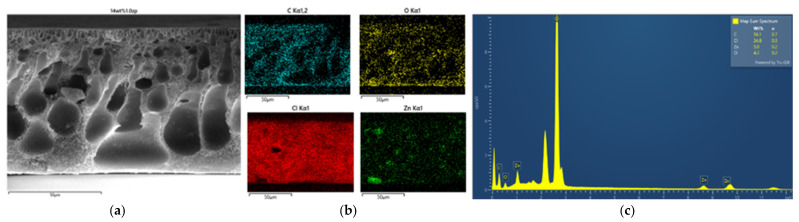
SEM image of PVC 14 wt% and 1.0 wt% of ZnO (**a**) cross-section, (**b**) mapping images, and (**c**) element spectrum.

**Figure 11 polymers-16-03551-f011:**
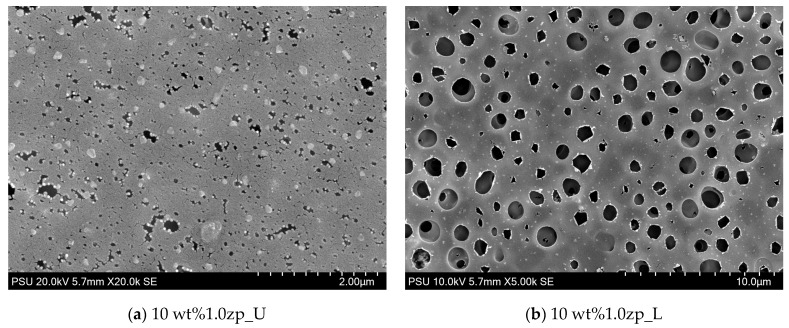
SEM image of PVC 10 wt% and 1.0 wt% of ZnO: (**a**) upper surface (not contact with glass plate) and (**b**) lower surface (contact with non-woven).

**Figure 12 polymers-16-03551-f012:**
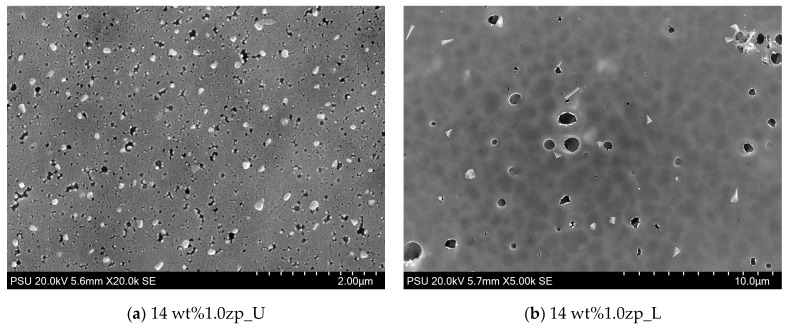
SEM image of PVC 14 wt% and 1.0 wt% of ZnO: (**a**) upper surface (not contact with glass plate) and (**b**) lower surface (contact with non-woven).

**Figure 13 polymers-16-03551-f013:**
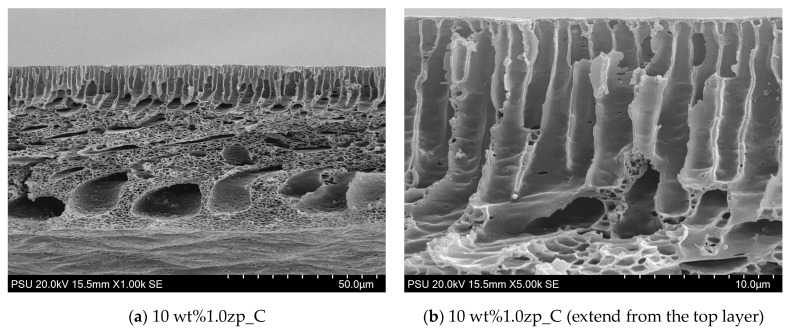
SEM image of PVC 10 wt% and 1.0 wt% of ZnO: (**a**) lower magnification at 1000 times, and (**b**) higher magnification at 5000 times.

**Figure 14 polymers-16-03551-f014:**
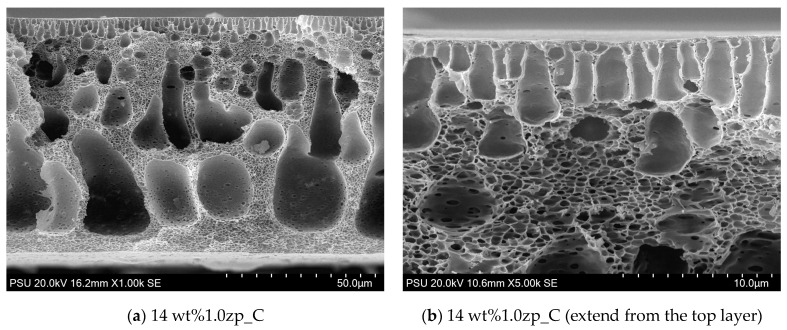
SEM image of PVC 14 wt% and 1.0 wt% of ZnO: (**a**) lower magnification at 1000 times, and (**b**) higher magnification at 5000 times.

**Figure 15 polymers-16-03551-f015:**
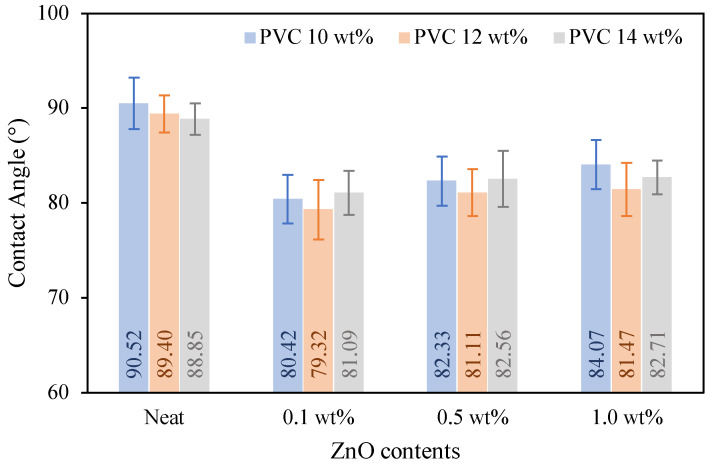
The wettability of PVC and its composite membranes depending on ZnO NP content and PVC loading.

**Figure 16 polymers-16-03551-f016:**
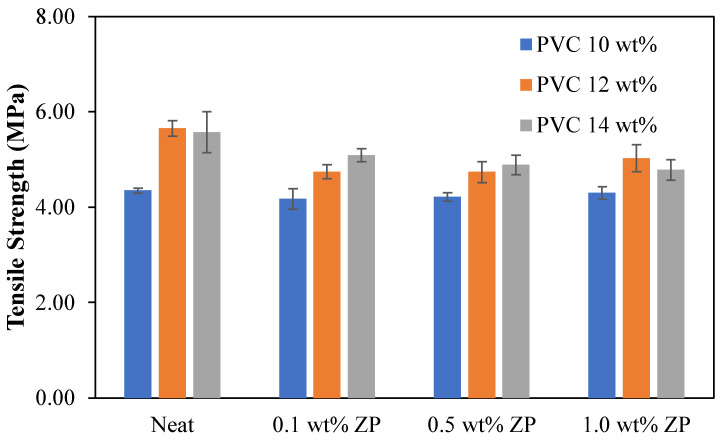
Tensile strength of PVC-based membranes with a different concentration of ZnO particles.

**Figure 17 polymers-16-03551-f017:**
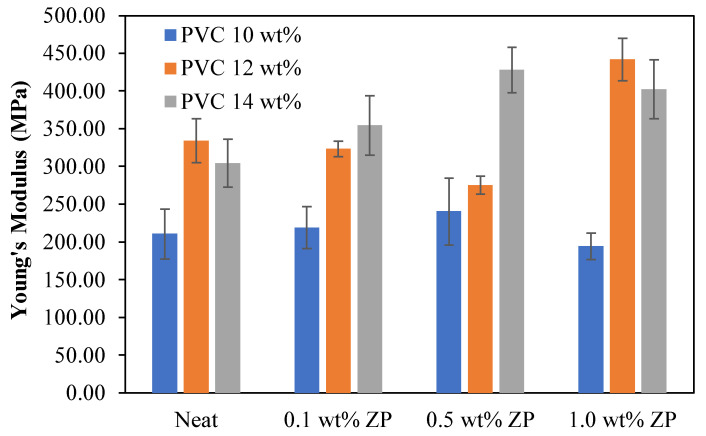
Young’s modulus of PVC-based membranes with a different concentration of ZnO particles.

**Figure 18 polymers-16-03551-f018:**
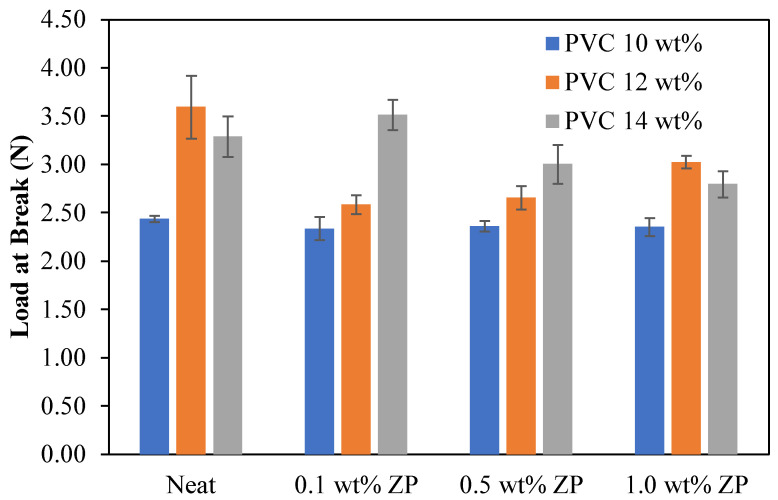
Load at break of PVC-based membranes with a different concentration of ZnO particles.

**Figure 19 polymers-16-03551-f019:**
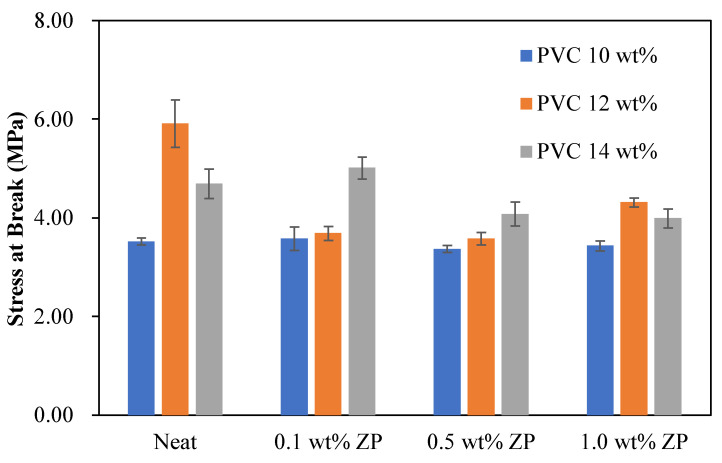
Stress at break of PVC-based membranes with a different concentration of ZnO particles.

**Figure 20 polymers-16-03551-f020:**
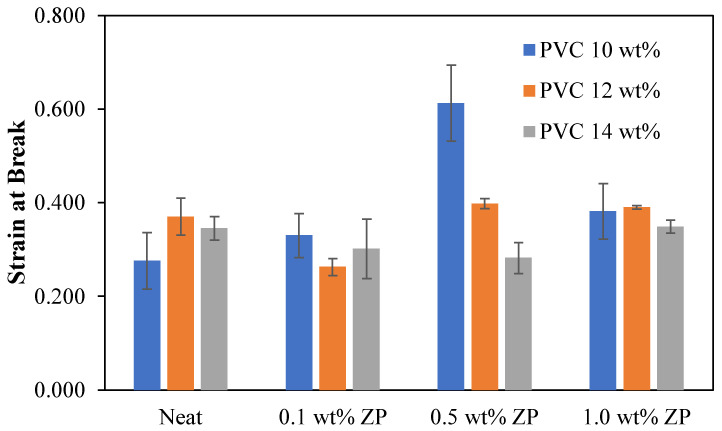
Strain at break of PVC-based membranes with a different concentration of ZnO particles.

**Figure 21 polymers-16-03551-f021:**
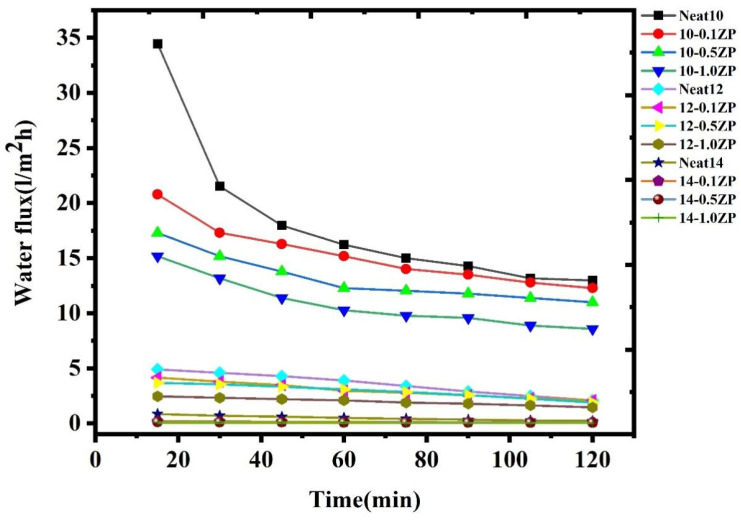
The PWF of the neat and composite PVC-ZnO membranes.

**Figure 22 polymers-16-03551-f022:**
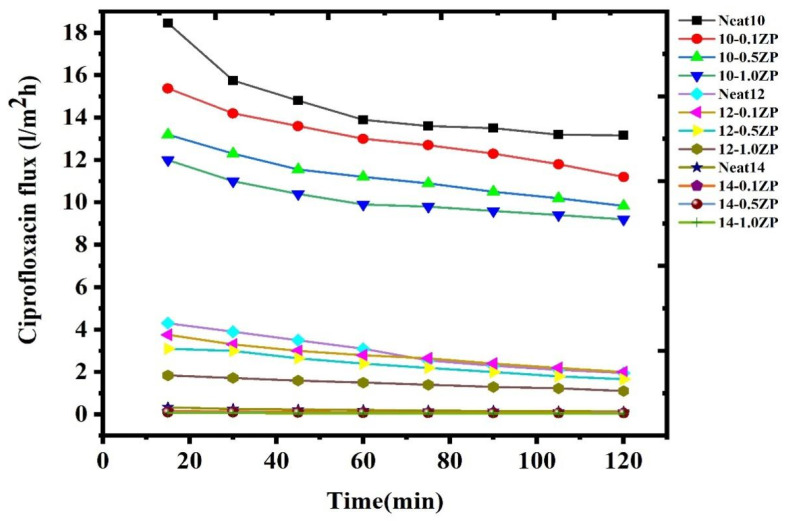
Flux for ciprofloxacin without water PVA polymer by PVC-ZnO composite membranes.

**Figure 23 polymers-16-03551-f023:**
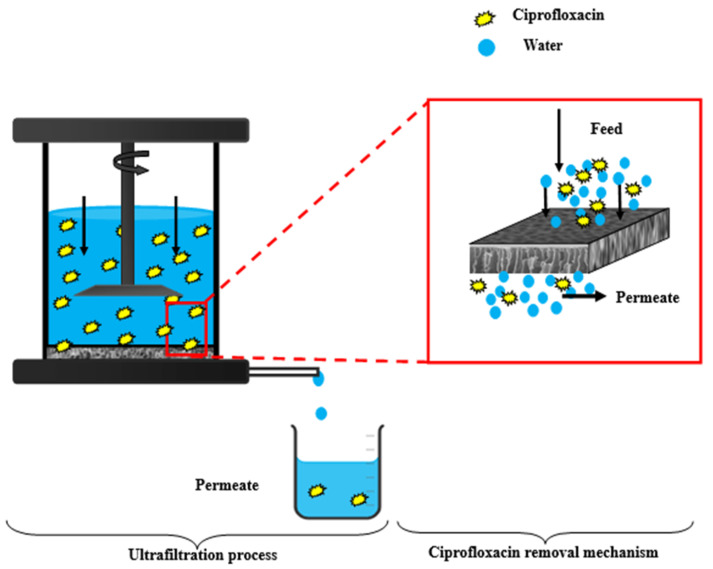
Schematic of the ultrafiltration process and ciprofloxacin removal mechanism.

**Figure 24 polymers-16-03551-f024:**
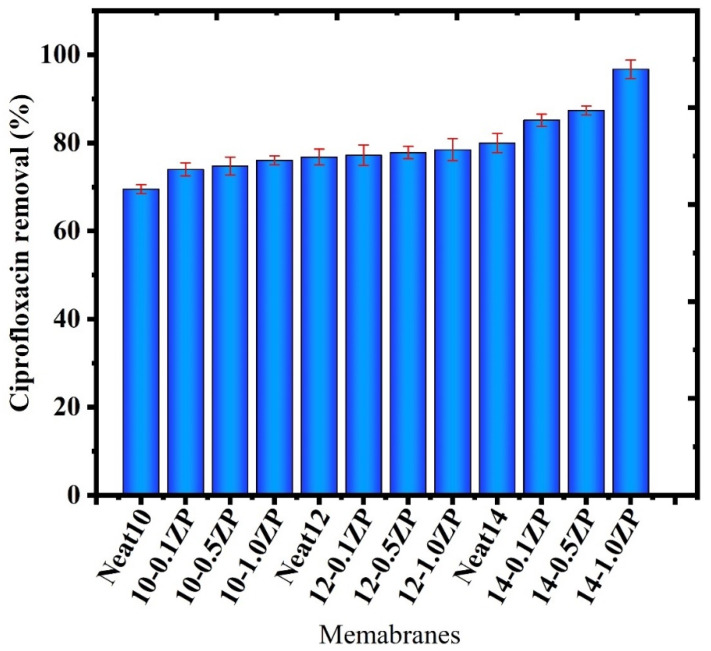
Rejection percentage of ciprofloxacin removal without PVA polymer.

**Figure 25 polymers-16-03551-f025:**
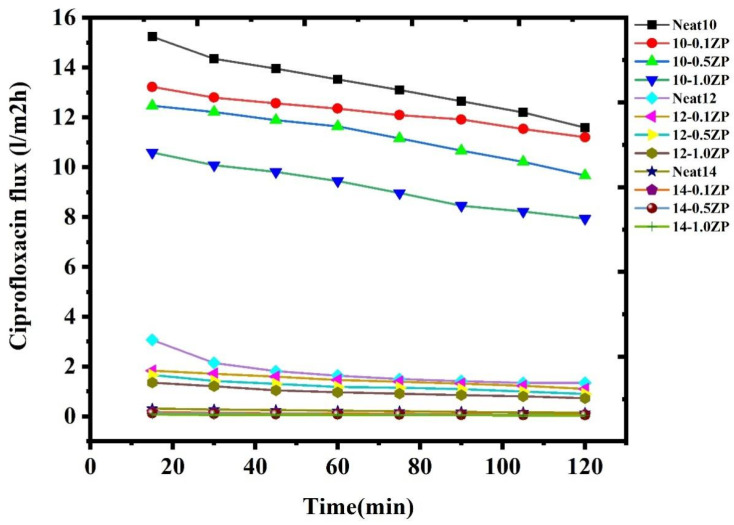
Flux for ciprofloxacin with PVA polymer by PVC-ZnO composite membranes.

**Figure 26 polymers-16-03551-f026:**
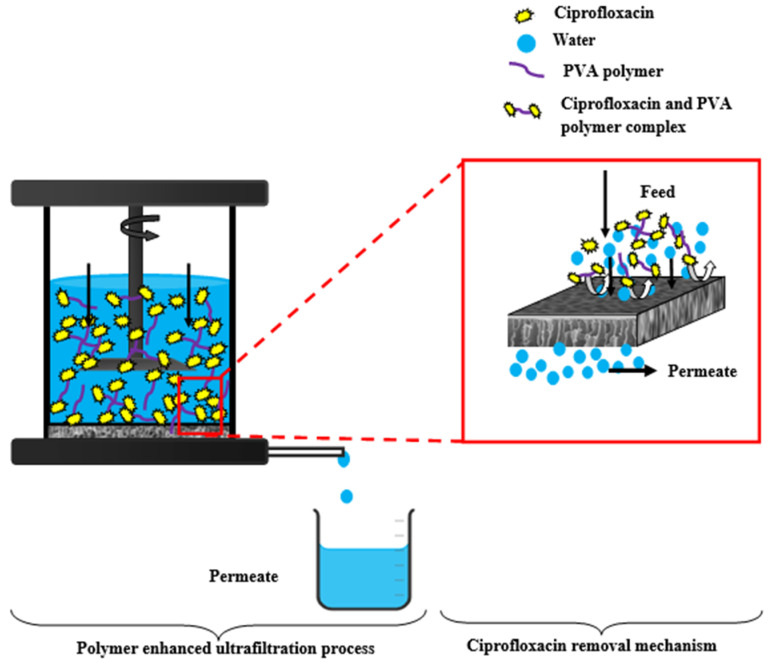
Schematic of the polymer-enhanced ultrafiltration process and ciprofloxacin removal mechanism.

**Figure 27 polymers-16-03551-f027:**
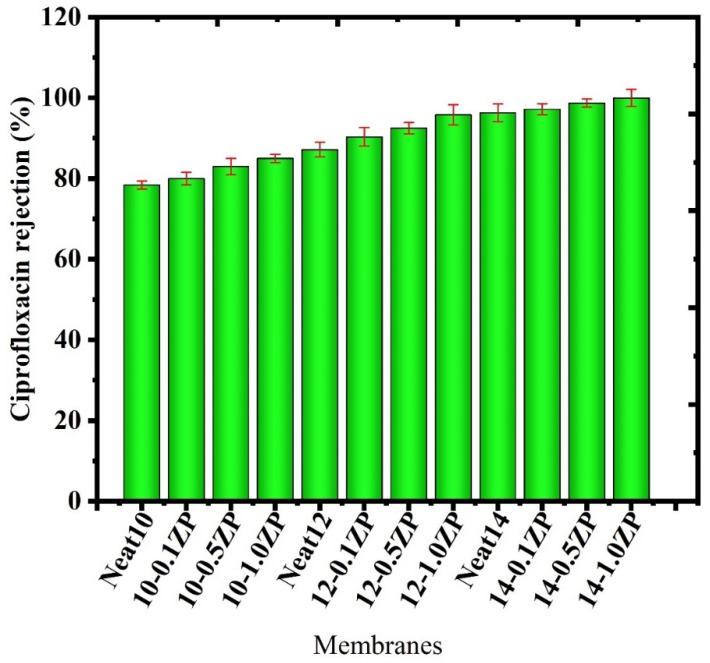
Rejection percentage of ciprofloxacin removal with PVA polymer.

**Table 1 polymers-16-03551-t001:** Composition of the prepared dope solutions.

Membrane Name	PVC (g)	ZnO NPs (g)	DMF (g)
**N10**	10	-	90
**N12**	12	-	88
**N14**	14	-	86
**N10-0.1ZP**	10	0.1	90
**N10-0.5ZP**	10	0.5	90
**N10-1.0ZP**	10	1.0	90
**N10-3.0ZP**	10	3.0	87

**Table 2 polymers-16-03551-t002:** Element percentages of mapping in each condition.

Condition	Area	Carbon		Chlorine		Oxygen		Zinc	
		wt % ^a^	at % ^b^	wt % ^a^	at % ^b^	wt % ^a^	at % ^b^	wt % ^a^	at % ^b^
N14	Surface	77.6	90.31	20.6	8.12	1.8	1.57	-	-
N14	Cross-section	69.9	85.55	26.0	10.78	4.0	3.67	-	-
N10-1.0ZP	Surface	64.1	84.01	28.5	12.65	2.1	2.07	5.3	1.28
N14-1.0ZP	Surface	84.9	93.72	10.3	3.85	2.3	1.91	2.6	0.53
N10-1.0ZP	Cross-section	57.1	77.44	28.3	13.00	7.7	7.84	6.9	1.72
N14-1.0ZP	Cross-section	56.1	77.96	34.8	16.38	4.2	4.38	5.0	1.28

Note: ^a^ The weight % data were collected from EDX at the top surface and cross-section of the flat sheet membrane. ^b^ The atomic weight was calculated through the conversion of weight % to atomic %.

**Table 3 polymers-16-03551-t003:** The porosity, water uptake, and overall mean pore size of PVC/ZnO composite membranes.

PVC/ZnO (wt%)	Porosity (%)	Water Uptake (%)	$\mathrm{Overall}\;\mathrm{Mean}\;\mathrm{Pore}\;\mathrm{Size}\;\left(\mu m\right)$
Neat 10	73.20 ± 3.12	67.77 ± 0.35	8.99
10-0.1ZP	75.87 ± 2.18	70.56 ± 0.15	10.83
10-0.5ZP	76.19 ± 0.98	69.62 ± 0.11	22.36
10-1.0ZP	75.24 ± 0.96	67.24 ± 0.11	24.13
Neat 12	76.84 ± 1.18	70.50 ± 0.23	5.91
12-0.1ZP	79.03 ± 0.62	70.63 ± 0.10	10.93
12-0.5ZP	77.81 ± 1.27	69.95 ± 0.04	22.41
12-1.0ZP	77.20 ± 1.09	68.46 ± 0.56	24.23
Neat 14	78.60 ± 1.39	70.68 ± 0.34	5.76
14-0.1ZP	80.56 ± 0.83	71.46 ± 0.17	11.15
14-0.5ZP	79.36 ± 0.84	71.50 ± 0.24	21.95
14-1.0ZP	78.58 ± 0.20	70.47 ± 0.12	23.51

**Table 4 polymers-16-03551-t004:** The mechanical properties of PVC are 10.0 wt% with the addition of ZnO particles.

	Tensile Strength (MPa)	Young’s Modulus (MPa)	Load at Break (N)	Stress at Break (MPa)	Strain at Break (-)
**N10**	4.35 ± 0.06	210.77 ± 33.23	2.43 ± 0.03	3.52 ± 0.07	0.276 ± 0.039 ^a^
**N10-0.1ZP**	4.17 ± 0.21	219.02 ± 27.74	2.34 ± 0.12	3.59 ± 0.24	0.330 ± 0.047 ^ab^
**N10-0.5ZP**	4.22 ± 0.09	240.49 ± 44.27	2.36 ± 0.05	3.37 ± 0.07	0.613 ± 0.081 ^c^
**N10-1.0ZP**	4.30 ± 0.13	194.33 ± 28.13	2.35 ± 0.09	3.44 ± 0.10	0.382 ± 0.059 ^b^
**% C.V.**	6.58	219.04	5.26	7.45	9.97
**F-test**	ns	ns	ns	ns	**

**Note** (1) Data are evaluated for all three readers and are summarized as mean ± standard deviation. (2) Different superscript letters indicate a significant difference (*p* < 0.01) compared with the least square difference (LSD) method. (3) ns = nonsignificant (*p* > 0.10), * = significant (*p* < 0.05), and ** = significant (*p* < 0.01) %C.V. = Coefficient of Variation.

**Table 5 polymers-16-03551-t005:** The mechanical properties of PVC are 12 wt% with the addition of ZnO particles.

	Tensile Strength (MPa)	Young’s Modulus (MPa)	Load at Break (N)	Stress at Break (MPa)	Strain at Break (-)
**N12**	5.65 ± 0.16 ^c^	334.31 ± 29.33 ^b^	3.59 ± 0.32 ^d^	5.91 ± 0.48 ^d^	0.371 ± 0.039 ^c^
**N12-0.1ZP**	4.74 ± 0.15 ^b^	323.53 ± 10.17 ^b^	2.58 ± 0.10 ^b^	3.69 ± 0.14 ^b^	0.263 ± 0.018 ^b^
**N12-0.5ZP**	4.74 ± 0.22 ^b^	275.21 ± 11.89 ^a^	2.65 ± 0.12 ^b^	3.58 ± 0.13 ^b^	0.398 ± 0.011 ^c^
**N12-1.0ZP**	5.03 ± 0.29 ^b^	441.76 ± 28.13 ^c^	3.02 ± 0.06 ^c^	4.32 ± 0.09 ^c^	0.390 ± 0.004 ^c^
**% C.V.**	9.56	110.36	10.25	12.61	3.72
**F-test**	**	**	**	**	**

**Note** (1) Data are evaluated for all three readers and are summarized as mean ± standard deviation. (2) Different superscript letters indicate a significant difference (*p* < 0.01) compared with the least square difference (LSD) method. (3) ns = nonsignificant (*p* > 0.10), * = significant (*p* < 0.05), and **= significant (*p* < 0.01) %C.V. = Coefficient of Variation.

**Table 6 polymers-16-03551-t006:** The mechanical properties of PVC are 14 wt% with the addition of ZnO particles.

	Tensile Strength (MPa)	Young’s Modulus (MPa)	Load at Break (N)	Stress at Break (MPa)	Strain at Break (-)
**N14**	5.57 ± 0.43 ^b^	304.58 ± 31.60 ^a^	3.29 ± 0.21 ^b^	4.70 ± 0.30 ^b^	0.346 ± 0.025 ^c^
**N14-0.1ZP**	5.09 ± 0.14 ^a^	354.60 ± 39.21 ^b^	3.51 ± 0.16 ^b^	5.02 ± 0.22 ^b^	0.301 ± 0.063 ^ac^
**N14-0.5ZP**	4.89 ± 0.20 ^a^	428.28 ± 30.18 ^c^	3.00 ± 0.20 ^a^	4.08 ± 0.25 ^a^	0.282 ± 0.033 ^ab^
**N14-1.0ZP**	4.78 ± 0.21 ^a^	402.45 ± 39.24 ^c^	2.80 ± 0.14 ^a^	3.99 ± 0.19 ^a^	0.349 ± 0.014 ^ac^
**% C.V.**	11.93	182.94	10.04	11.53	6.81
**F-test**	**	**	**	**	*

**Note** (1) Data are evaluated for all three readers and are summarized as mean ± standard deviation. (2) Different superscript letters indicate a significant difference (*p* < 0.01) compared with the least square difference (LSD) method. (3) ns = nonsignificant (*p* > 0.10), * = significant (*p* < 0.05), and **= significant (*p* < 0.01) %C.V. = Coefficient of Variation.

## Data Availability

The data presented in this work are available upon request from the corresponding authors.

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
