# Peer review of "Fabrication and Performance Evaluation of a Novel Composite PVC-ZnO Membrane for Ciprofloxacin Removal by Polymer-Enhanced Ultrafiltration"

_polymers, 2024, doi:10.3390/polym16243551_

Round 1

Reviewer 1 Report

Comments and Suggestions for Authors

Please see attached my recommendations for the authors

Comments on the Quality of English Language

It can improve!

Reviewer 2 Report

Comments and Suggestions for Authors

In this article, the Polymer Enhanced Ultrafiltration method, which is predicated on the creation of a composite PVC-ZnO membrane, was used to eradicate ciprofloxacin. The characterizations of obtained membrane are enough and discussion and explanation are reasonable. However, some details of experiment are needed to improve. Therefore, minor revisions are recommended before it is accepted for publication. The detailed comments are listed below:

1.     Other antibiotics should be used to test membrane performance.

2.     Figure 4 is confused, and improve it.

3.     Why the flux is so low. Please give the explanation. What is the advantage of this membrane comparing to other membranes in literatures.

4.     (line 602-606) Please give the explanation why the flux decreases with the increase of concentration.

5.     (Figure 11-14) The effect of ZnO concentration on membrane morphology should be explained.

6.     The language need be improved.

Comments on the Quality of English Language

The language need be improved.

Round 2

Reviewer 1 Report

Comments and Suggestions for Authors

The paper has improved in line to reviewer comments so that it can be accepted for publication